# Variability for Nitrogen Management in Genetically-Distant Maize (*Zea mays* L.) Lines: Impact of Post-Silking Nitrogen Limiting Conditions

**Isabelle Quilleré [1], Céline Dargel-Graffin [1], Peter J. Lea [2] and Bertrand Hirel [1,\*]**

[1]   Institut Jean-Pierre Bourgin, INRA, Agro-ParisTech, Université de Paris-Saclay, 78000 Versailles, France; isabelle.quillere@inra.fr (I.Q.); celine.dargel-graffin@inra.fr (C.D.-G.)

[2]   Lancaster Environment Centre, Lancaster University, Lancaster LA1 4YQ, UK; p.lea@lancaster.ac.uk

[\*]   Correspondence: bertrand.hirel@inra.fr; Tel.: +33-1-30-83-30-89

**Abstract:** The impact of nitrogen (N)-limiting conditions after silking on kernel yield (KY)-related traits and whole plant N management was investigated using fifteen maize lines representative of plant genetic diversity in Europe and America. A large level of genetic variability of these traits was observed in the different lines when post-silking fertilization of N was strongly reduced. Under such N-fertilization conditions, four different groups of lines were identified on the basis of KY and kernel N content. Although the pattern of N management, including N uptake and N use was variable in the four groups of lines, a number of them were able to maintain both a high yield and a high kernel N content by increasing shoot N remobilization. No obvious relationship between the genetic background of the lines and their mode of N management was found. When N was limiting after silking, N remobilization appeared to be a good predictive marker for identifying maize lines that were able to maintain a high yield and a high kernel N content irrespective of their female flowering date. The use of N remobilization as a trait to select maize genotypes adapted to low N input is discussed.

**Keywords:** genetic diversity; maize; nitrogen; remobilization; silking; uptake; [15]N-labeling

## 1. Introduction

The application of mineral nitrogen (N) fertilizers is one of the main agricultural practices used to maintain and restore soil fertility. It is able to stabilize or even increase yield for the majority of crop plants, including cereals such as maize. The applied mineral N is particularly soluble for easy uptake by plants, allowing the rapid assimilation of N during root and shoot vegetative growth [1], and thus ensuring the production of food for the constantly-growing world population [2].

Consequently, there has been, over the past 70 years, an almost five-fold increase in the total N applied to crops. In contrast, in harvestable material, such as grains used for human food and animal feed, the protein content has only increased by a factor of 3. This indicates that there was a 30% decrease in N-use efficiency (NUE), which can be defined as the yield obtained/unit of available N in the soil (supplied by the soil + N fertilizer) [3].

Plant NUE is the product of N-uptake efficiency (amount of N taken up/quantity of available N) and of the N-utilization efficiency (yield/absorbed N), [4]. There is large genetic variability of both N-uptake efficiency and N-utilization efficiency in many crops, notably in maize [5,6]. However, when examining the contribution of these two biological processes to the overall plant NUE, it has often been observed that the best performing maize varieties at high N fertilization input are not the best ones when the N fertilizer supply is lowered [7,8]. Such poor of performance under low N fertilization

input is partly due to the occurrence of interactions between the genotype and the level of N present in the soil, but notably because most of the previous breeding strategies have been conducted under non-limiting N fertilization conditions. Therefore, the opportunities of selecting for high-yielding maize varieties when the N fertilization conditions are low have been missed, or not fully exploited [5].

Although it has been shown that maize productivity can be maintained under low-N input [9], high N fertilization rates have been, and are still, used in most high-yielding intensive agricultural maize production systems [10] and in breeding strategies [11,12]. However, under such high N fertilization inputs, over 50% and up to 75% of the mineral N applied to the field is not taken up by the plant and is lost by leaching into the soil [13]. Such a N loss leads to the eutrophication of freshwater and marine ecosystems [14–16] and to the emissions of $N_2O$ (nitrous oxide), which has a global warming potential almost 300 times that of $CO_2$ [17]. The chemical synthesis of N fertilizers also increases crop production costs [18,19]. Altogether, both energy input for N fertilizer production and NUE are considered to be important indicators for the environmental impact of the production of most conventional food and energy crops such as maize [20].

It is estimated that an increase in agricultural production by at least 70%, will be necessary in order to feed the 9 billion people projected for the world population in 2050 [21]. Therefore, developing more sustainable agricultural practices based on fertilizer use rationalization and selecting or producing genetically-engineered genotypes exhibiting improved NUE [22,23] are possible ways of overcoming the detrimental impact of the overuse of N fertilizers [1,6].

In maize, 45–65% of the grain N, which acts a source of proteins for both humans and animals, is provided from pre-existing N in the stover before silking. Nitrogen translocation from the stover to the grain is very dependent upon the environmental conditions and/or the genotype [7]. The remaining 35–55% of the grain N originates from post-silking N uptake [7,24,25]. Therefore, to identify or select maize genotypes that exhibit improved N uptake before and after silking, it will be necessary to improve our understanding of the physiological and genetic determinants that govern these two biological processes. Such an improved understanding can be obtained using lines or hybrids for which their relative contribution is variable, and then proposing strategies to provide N when the plant needs it most.

In previous studies, remarkably large genetic variability was observed in the leaf metabolite content, leaf enzyme activities, leaf biomass-related components and KY of a core collection of maize lines [26], including races originating from different northern and southern countries of both America and Europe [27]. Therefore, in the present study, we have exploited the large genetic variability of these maize lines to examine the impact of N deprivation after silking on plant N uptake and plant productivity. Only a limited number of studies were undertaken to determine whether compensatory mechanisms such as N remobilization occur in genetically-distant maize lines when there is reduction in N availability during the grain filling period [7,24,25]. Whether these compensatory mechanisms could be used to select maize lines with improved NUE when there is a shortage of N after silking is discussed.

## 2. Materials and Methods

### 2.1. Plant Material

Seeds of the 15 maize inbred lines selected for the experiment were obtained from the core collection used for association genetics studies [27,28] of the Institut National de la Recherche Agronomique (INRA), Saint-Martin-de-Hinx, France. These inbred lines were classified into five main maize groups: Tropical (2 lines: EML1201 and P465P), European Flint (3 lines: Lo3, Lo32 and FV2), Northern Flint (3 lines: NYS302, C105 and ND36), Corn Belt Dent (6 lines: ND283, Mo17, FV252, SA24U, MBS847 and HP301) and Stiff Stalk (1 line: B73) [27]. This original classification was organized on the basis of genome sequence polymorphism of lines using Simple Sequence Repeat (SSR) microsatellite markers, and later on using Single Nucleotide Polymorphism (SNP) markers as

previously described [27,29]. The group named Tropical contained lines from Argentina, Mexico and Spain, whereas the Corn Belt Dent lines mostly originated from North America and contained two Popcorn lines (HP301 and SA24). Line FV252 also belonged to the Corn Belt Dent lines but originated from France. In the lines classified as European Flint, there were lines from France and Italy, whereas all the Northern Flint lines were from North America. The 15 maize lines were grown in a glasshouse at INRA, Versailles, France (N 48°48.133′, E 2°04.942′), until maturity without any additional light or heat from April 5th to September 19th, 2012. Seeds were first sown on coarse sand and after 2 weeks, when 2 to 3 leaves had emerged, 8 individual seedlings of each line of a similar height were transferred to pots for $^{15}$N-labeling experiments and yield measurements. Each plant was transferred to a separate pot (diameter and height of 22 cm, volume 7l) containing clay loam soil and grown until maturity in the glasshouse. Clay loam soil was composed of a mixture of loam (washed fine silt with no minerals) and loam balls of ~0.5 cm diameter that ensured sufficient aeration of the roots and allowed the growth of the plant until maturity without lodging. Clay loam soil also allowed a constant flow of the provided nutrient solution.

All plants were watered four times a day with a complete nutrient solution containing 10 mM N (8 mM $NO_3^-$ + 2 mM $NH_4^+$) [30]. The complete nutrient solution ($N^+$) contained 5 mM $K^+$, 3 mM $Ca^{2+}$, 0.4 mM $Mg^{2+}$, 1.1 mM $H_2PO_4^-$, 1 mM $SO_4^{2-}$, 1.1 mM $Cl^-$ 21.5 μM $Fe^{2+}$ (Sequestrene; Ciba-Geigy, Basel, Switzerland), 23 μM $B^{3+}$, 9 μM $Mn^{2+}$, 0.3 μM $Mo^{2+}$, 0.95 μM $Cu^{2+}$ and 3.5 μM $Zn^{2+}$. The variation in the silking date between the 15 lines was approximately 5 weeks, starting from June 4th, 2012.

$^{15}$N-labeling was carried out on May 30th during vegetative growth, at the beginning of stem elongation (8 visible leaves, Biologishe Bundesanstalt Bunderssortenamt und Chemicshe Industrie (BBCH) 17) of the lines exhibiting the earliest development. Into each pot, 150 mL of a solution of 30 mM $KNO_3^-$ containing 2% $^{15}$N atom excess was applied to each pot in order to distribute on average 1.25 mg $^{15}$N per plant. After silking, the plants were separated into 8 groups of 1 plant for each of the 15 lines. Four groups were watered with a complete nutrient solution ($N^+$) and the other four groups with a low N ($N^-$) solution. The eight groups were randomly distributed in the greenhouse to ensure homogenous plant growth irrespective of the line and the N treatment. The composition of the low N solution ($N^-$) was similar to that of the complete nutrient solution except that 8 mM $NO_3^-$ + 2 mM $NH_4^+$, 1 mM $SO_4^{2-}$ and 1.1 mM $Cl^-$ were replaced by 0.3 mM $NO_3^-$, 2.2 mM $SO_4^{2-}$ and 2.3 mM $Cl^-$ respectively. Kernel yield (KY), kernel number (KN), thousand kernel weight (TKW) and kernel N content (%NK) were determined according to the methods described in [31] using individual plants from $N^+$ and in $N^-$ in each of the 8 groups, making four replicates in total for each of the two N feeding conditions.

## 2.2. Determination of N Content and $^{15}$N-Abundance

Plants were harvested when all the kernels were matured. Plant samples were separated in two different batches one corresponding to the shoots (leaves + stalk + sheaths + husk = SDW (Shoot Dry Weight)) and the other to the ear (cob + kernels). After the drying (70 °C in an oven) and weighing of shoots and kernels, the material was ground to obtain a homogeneous fine powder. A sub-sample of 1 mg was used to determine total N content and $^{15}$N-abundance by an elemental automated analyser (Roboprep CN, SERCON Europa Scientific Ltd, Crewe, UK) coupled to an isotope ratio mass spectrometer (Tracermass, PDZ Europa Scientific Ltd, Crewe, UK) calibrated for measuring $^{15}$N-natural abundance. As the amount of N present in the cobs is very low at plant maturity [32], it was not considered in calculating the plant N budget.

$^{15}$N abundance was calculated as atom per cent (A%), and defined as A% = 100 × ($^{15}$N) ($^{15}$N + $^{14}$N), both in labeled plant samples and in unlabeled control plants. A% in the unlabeled control plants was close to 0.36634%, a value which corresponds to the natural abundance of atmospheric dinitrogen ($N_2$). $^{15}$N enrichment (E%) of the plant samples was then defined as (A% sample − A% control). The amount of $^{15}$N contained in the sample (Q) was calculated on a dry weight (DW) basis using the following formula: Q = DW × E% × N% with N% being the concentration of N in

the sample. The amount of $^{15}$N present in the kernels compared to that present in the whole plant (shoots + ear) was calculated using the following formula: Q kernels/ (Q kernels + Q shoots) and named $^{15}$N-Harvest-Index ($^{15}$NHI). The $^{15}$NHI represents the proportion of $^{15}$N that is absorbed at the vegetative stage and further remobilized to the kernels at the reproductive stage after silking. Roots were not considered to calculate the $^{15}$NHI, since in previous studies [33], it was shown that under experimental conditions such as those employed in the present study the amount of $^{15}$N present in the roots represents less than 5% of the total $^{15}$N present in the whole plant.

*2.3. Statistical Analysis*

Statistical analysis of data was performed using the Student *t*-test functions of the XLStat-Pro 7.5 software, 2013 (Addinsoft, Bordeaux, France). Pearson correlation coefficients between yield-related traits, N accumulation before silking and the amount of N remobilized from the shoots to the kernels were calculated in order to identify their possible relationship. Yield, yield components, biomass accumulation and the N budget of the 15 lines grown in the two contrasting N feeding conditions, are presented as mean values for 4 plants with standard errors (SE) (SE = SD/$\sqrt{n}$), where SD is the standard deviation and *n* is the number of samples).

**3. Results**

*3.1. Plant Agronomic Performances*

Variations in KY in the glasshouse ranged from 27.3 g for line Lo32 to 105.8 g for line MBS847, with a mean of 67.3 g for all the lines. In previous field experiments [26], the variations in KY ranged from 32.5 g for line ND36 to 75.9 g for line MBS847, with a mean of 47 g for all the lines. The 15 lines grown in the glasshouse had a higher yield compared to those grown in the field, (40% higher) on average, explained by a higher KN and a higher TKW (20% on average). Correlation studies showed that between the glasshouse and the field experiment, there was a strong correlation of 0.8 (*p*-value < 0.0001) for KN and a good correlation of 0.63 (*p*-value < 0.01) for KY. In high-yielding lines (H) KY was >50 g and up to 120 g per plant and in low-yielding lines (L) KY was <50 g and down to 30 g per plant. Such a finding indicates that plant growth conditions were optimal in the glasshouse. Among the 15 lines examined, a large genetic variability was observed for the traits related to yield and plant N management. When a two-way ANOVA was performed, we found a significant genotypic effect (*p* < 0.0001) for all the measured traits. The variation in KY between the lowest and the highest yielding lines when the plants were grown either in N$^+$ or in N$^-$ was up to 4-fold. When a comparison was made between N$^-$ or N$^+$ conditions, two groups of lines could be identified on the basis of kernel production (Figure 1). One group did not show any significant reduction in KY in N$^-$, whereas in the other group, a decrease in KY ranging from 20% in line FV252 to 80% in line B73 was observed (Table 1). Interestingly, following growth in N$^-$, KY was not reduced in the two tropical lines EML1201 and P465P. For the Corn Belt, European Flint, Northern Flint lines and Stiff Stalk lines there was no clear relationship between their genetic characteristics and kernel production in N$^-$. As shown in Table 1, the decrease in KY was due to a decrease in KN, whereas the reduction in N fertilization after silking did not have any marked impact on TKW. In agreement with a number of previous reports on maize kernel production and its genetic variability [34], our data showed that in N$^+$ there was a positive correlation between KY and KN (*r* = 0.75, *p* < 0.001), whereas KY or KN and TKW were not correlated (*r* =0.42, *p* < 0.115 and *r* = 0.26, *p* < 0.359 respectively).

Following the reduction in post-silking N fertilization, four groups of lines were identified when kernel N content (%NK) was considered. The first group was represented by four lines (EML201, Lo3, NYS302 and ND283), which exhibited no change in KY and %NK. The second group (lines P465P, C105, Mo17, Lo32) exhibited on average a 10–15% decrease in kernel N and a concomitant decrease in KY. In the third group, there were six lines (FV252, ND36, SA24U, MBS847, HP301 and B73), which

exhibited a decrease in KY and no change in the %NK. Only line FV2 was representative of group 4, in which there was a marked decrease of KY (71%) and an increase in % NK content of 19%.

There was no clear relationship between the four groups of lines defined in the present study on the basis on their agronomic performance in $N^-$ and the five genetic groups they belong to (European Flint, Northern Flint, Corn Belt Dent, Tropical and Stiff Stalk; See Figure 1). Nevertheless, in the group of lines exhibiting a decrease in KY with no change in %NK in $N^-$ (group 3 in Figure 1), there are four lines out of the six that belong the Corn Belt genetic group.

In most of the lines, total shoot biomass production at maturity (SDW) including stems, leaves, tassels and husks was unaffected following growth in $N^-$ compared to $N^+$, except for lines NYS302, C105 and FV252, for which a 30% decrease was observed.

*3.2. Plant Nitrogen Management*

It can be seen that compared to the $N^+$-grown plants, most of the lines showed a significant reduction in the amount of Nt under $N^-$ conditions, ranging from 15% in line Lo3 to 56% in line ND36.

We observed that there were no significant differences between the amount of total plant N (Nt) accumulated following growth in $N^-$ (Table 1) and the Nt before siliking in $N^+$. This indicates that negligible amounts of N were later taken up by the plants under post-silking N-limiting conditions. There was an almost three-fold range of variation for Nt in $N^-$ (Table 1).

The amount of N accumulated by the plants that was further translocated to the kernels via the remobilization process ($^{15}$N-Harvest index, $^{15}$NHI) was estimated using the $^{15}$N. In four lines (NYS302, P465P, C105 and ND36) an increase of up to 40% in the $^{15}$N kernel content was observed in $N^-$. Such an increase was reduced to 20% in lines EML1201, Lo3 and FV252. In other lines such as ND283, Mo17, Lo32, SA24U, MBS847 and HP301, low N fertilization after silking did not have any significant impact on shoot N remobilization. Only in lines B73 and FV2 were lower amounts of $^{15}$N (50%) translocated to the kernels, after the labeling period.

As shown in Figure 1, it was observed that neither the silking date nor KY had any direct relationship with the capacity of a line to take up or remobilize N. For example, only high yielding lines (H) such as EML1201 or low yielding lines (L) such as ND36 were able to remobilize more N under N limiting conditions after silking. We also observed that silking in line ND36 occurred much earlier compared to line EML1201.

**Table 1.** Agronomic performance and nitrogen management of the core collection of 15 maize inbred lines originating from Europe and America.

| Group | Line | KY (g) ± SE | | | KN ± SE | | | %NK ± SE | | | SDW (g) ± SE | | | Nt (mg) ± SE | | | ¹⁵NHI ± SE | | |
|---|---|---|---|---|---|---|---|---|---|---|---|---|---|---|---|---|---|---|---|
| | | N⁺ | N⁻ | t-test | N⁺ | N⁻ | t-test | N⁺ | N⁻ | t-test | N⁺ | N⁻ | t-test | N⁺ | N⁻ | t-test | N⁺ | N⁻ | t-test |
| 1 | EML1201 | 92.6 ± 3.2 | 80.5 ± 8.5 | ns | 266 ± 8.4 | 216.5 ± 15.5 | * | 2.06 ± 0.12 | 1.94 ± 0.16 | ns | 136.2 ± 6.6 | 141.8 ± 14.3 | ns | 4.35 ± 0.19 | 3.07 ± 0.09 | ** | 0.36 ± 0.01 | 0.47 ± 0.03 | * |
| 1 | Lo3 | 86.1 ± 5.6 | 85.8 ± 6.2 | ns | 465 ± 12.9 | 469.3 ± 22.6 | ns | 1.86 ± 0.06 | 1.78 ± 0.03 | ns | 106.4 ± 12.8 | 91.4 ± 9.0 | ns | 3.09 ± 0.19 | 2.62 ± 0.16 | (*) | 0.46 ± 0.03 | 0.55 ± 0.02 | * |
| 1 | NYS302 | 35.2 ± 4.0 | 29.8 ± 4.1 | ns | 238 ± 20.3 | 174.5 ± 11.5 | ns | 2.56 ± 0.13 | 2.36 ± 0.10 | ns | 133.3 ± 3.2 | 90.7 ± 7.9 | ** | 3.69 ± 0.28 | 1.94 ± 0.13 | * | 0.21 ± 0.04 | 0.36 ± 0.04 | (*) |
| 1 | ND283 | 34.1 ± 2.0 | 35.3 ± 1.5 | ns | 177.7 ± 10.5 | 197.8 ± 3.8 | ns | 2.14 ± 0.15 | 2.14 ± 0.20 | ns | 43.1 ± 1.5 | 48.9 ± 4.0 | ns | 1.19 ± 0.08 | 1.20 ± 0.12 | ns | 0.61 ± 0.07 | 0.67 ± 0.05 | ns |
| 2 | P465P | 56.7 ± 7.6 | 70.4 ± 1.3 | ns | 298.5 ± 60.1 | 311.7 ± 13.3 | ns | 2.33 ± 0.08 | 2.04 ± 0.10 | (*) | 155.4 ± 17.7 | 134.7 ± 14.3 | ns | 3.33 ± 0.22 | 2.44 ± 0.13 | * | 0.38 ± 0.08 | 0.61 ± 0.04 | (*) |
| 2 | C105 | 52.2 ± 4.1 | 62.3 ± 8.3 | ns | 209 ± 14.0 | 227.3 ± 28.6 | ns | 2.17 ± 0.08 | 1.85 ± 0.07 | * | 81.8 ± 3.0 | 61.7 ± 7.8 | * | 2.68 ± 0.11 | 1.87 ± 0.11 | ** | 0.37 ± 0.01 | 0.62 ± 0.04 | ** |
| 2 | Mo17 | 94.4 ± 5.5 | 87.9 ± 15.7 | ns | 298.1 ± 21.3 | 295.8 ± 46.6 | ns | 2.31 ± 0.04 | 1.99 ± 0.13 | (*) | 155.3 ± 11.0 | 144.9 ± 7.5 | ns | 4.35 ± 0.23 | 3.31 ± 0.09 | ** | 0.44 ± 0.03 | 0.47 ± 0.07 | ns |
| 2 | Lo32 | 27.3 ± 0.4 | 24.8 ± 6.4 | ns | 223 ± 15.0 | 189 ± 47.8 | ns | 2.39 ± 0.04 | 2.15 ± 0.04 | * | 126.9 ± 13.0 | 121.0 ± 18.1 | ns | 3.28 ± 0.49 | 2.27 ± 0.20 | ns | 0.17 ± 0.04 | 0.24 ± 0.08 | ns |
| 3 | FV252 | 76.3 ± 3.5 | 59.1 ± 5.4 | * | 372.2 ± 20.1 | 295.4 ± 25.8 | * | 2.04 ± 0.13 | 1.93 ± 0.10 | ns | 99.9 ± 6.7 | 80.0 ± 6.6 | (*) | 3.07 ± 0.08 | 1.98 ± 0.15 | *** | 0.48 ± 0.03 | 0.59 ± 0.04 | * |
| 3 | ND36 | 37.2 ± 3.2 | 24.5 ± 0.8 | (*) | 245.7 ± 15.0 | 152.3 ± 3.8 | * | 2.58 ± 0.11 | 2.38 ± 0.27 | ns | 99.1 ± 16.9 | 62.0 ± 8.6 | ns | 3.03 ± 0.40 | 1.32 ± 0.19 | (*) | 0.30 ± 0.03 | 0.48 ± 0.02 | * |
| 3 | SA24U | 122.5 ± 2.5 | 85.1 ± 8.6 | (*) | 793.3 ± 0.3 | 553.8 ± 81.3 | (*) | 1.97 ± 0.08 | 2.02 ± 0.04 | ns | 191.2 ± 1.5 | 126.3 ± 29.0 | ns | 4.73 ± 0.02 | 2.99 ± 0.34 | * | 0.43 ± 0.01 | 0.60 ± 0.14 | ns |
| 3 | MBS847 | 105.9 ± 8.7 | 59.9 ± 7.4 | ** | 543 ± 32.3 | 311.4 ± 41.3 | ** | 2.12 ± 0.08 | 2.15 ± 0.07 | ns | 119.3 ± 11.1 | 107.9 ± 12.3 | ns | 3.94 ± 0.31 | 2.56 ± 0.28 | * | 0.52 ± 0.04 | 0.49 ± 0.04 | ns |
| 3 | HP301 | 49.3 ± 4.9 | 34.1 ± 3.6 | (*) | 468.9 ± 49.5 | 306.3 ± 39.1 | (*) | 2.32 ± 0.15 | 2.02 ± 0.17 | ns | 130.6 ± 2.1 | 160.2 ± 18.4 | ns | 3.12 ± 0.12 | 2.50 ± 0.10 | * | 0.32 ± 0.00 | 0.26 ± 0.04 | ns |
| 3 | B73 | 85.4 ± 19.0 | 19.2 ± 3.0 | (*) | 391.5 ± 54.0 | 97.8 ± 4.3 | * | 2.29 ± 0.27 | 2.45 ± 0.22 | ns | 159.2 ± 12.7 | 156.5 ± 27.5 | ns | 4.38 ± 0.14 | 3.03 ± 0.72 | (*) | 0.38 ± 0.05 | 0.16 ± 0.03 | * |
| 4 | FV2 | 54.4 ± 5.7 | 15.8 ± 2.6 | ** | 258.6 ± 19.8 | 76.7 ± 10.3 | *** | 2.04 ± 0.06 | 2.42 ± 0.16 | * | 75.5 ± 7.3 | 71.8 ± 7.0 | ns | 2.23 ± 0.20 | 1.55 ± 0.20 | (*) | 0.48 ± 0.04 | 0.24 ± 0.05 | * |

KY = Kernel Yield; KN = Kernel Number; %NK = Kernel nitrogen content; SDW = Shoot Dry Weight (leaves + stalk + sheaths + husk); Nt = Total Nitrogen in the whole plant (shoots + ear); ¹⁵NHI = N-Havest index. N⁺ = Nitrogen non-limiting conditions. N⁻ = Post-silking N-limiting conditions. Group = Four groups of lines identified on the basis of KY and on the %NK (see also Figure 1). Asterisks indicate a *t*-test with a *p*-value: (*) $p < 0.10$; * $p < 0.05$; ** $p < 0.01$; *** $p < 0.001$; ns, nonsignificant.

| Line | KY | %N K | Group | NUp | NRem | Class | KYL N$^+$ | KYL N$^-$ | Sd |
|---|---|---|---|---|---|---|---|---|---|
| EML1201 | = | = | 1 | - | + | N-tolerant | H | H | M |
| Lo3 | = | = | 1 | - | + | | H | H | M |
| NYS302 | = | = | 1 | - | + | | L | L | M |
| ND283 | = | = | 1 | = | = | | L | L | E |
| P465P | = | - | 2 | - | + | N-semi tolerant | H | H | M |
| C105 | = | - | 2 | - | + | | H | H | M |
| Mo17 | = | - | 2 | - | = | | H | H | L |
| Lo32 | = | - | 2 | = | = | | L | L | M |
| FV252 | - | = | 3 | - | + | N-semi sensitive | H | H | E |
| ND36 | - | = | 3 | - | + | | L | L | E |
| SA24U | - | = | 3 | - | = | | H | H | L |
| MBS847 | - | = | 3 | - | = | | H | H | M |
| HP301 | - | = | 3 | - | = | | L | L | L |
| B73 | - | = | 3 | - | - | | H | L | L |
| FV2 | - | + | 4 | - | - | N-sensitive | H | L | E |

**Figure 1.** Schematic representation of the distribution of the 15 maize inbred maize lines originating from Europe and America in relation to their agronomic performance and their mode of N management under post-silking N-limiting conditions. On the left of the panel (Line), colored boxes represent the five groups of maize lines obtained from different countries of Europe and America (Tropical: pale green; European Flint: Blue; Northern Flint: dark green; Corn Belt Dent: yellow and Stiff Stalk: orange). Kernel yield (KY), kernel number (KN) and kernel N content (%NK) were determined for the 15 maize lines grown in the glass house (Table 1). For the different measured traits including KY, %NK, N remobilization (NRem) and N uptake (NUp) the red background color indicates an increase (+), the green background color a decrease (-) and the blue background color no change (=) under N-limiting conditions after silking until plant maturity. Groups 1, 2, 3 and 4 represent the four groups of lines identified on the basis of KY and %NK. Class corresponds to a proposed ranking of the lines based on their N management response when post-silking N fertilization is reduced. On the right of the panel, the period for the silking date (Sd) is shown. E = early (from 4th to the 18th of June); M = medium (from the 19th to the 29th of June); L = late (from the 30th of June to the 6th of July), along with the level of kernel yield in N$^+$ (KYL N$^+$) and in N$^-$ (KYL N$^-$); H = High > 50 g and up to 120 g per plant; L= low < 50 g and down to 30g per plant. Detailed yield data are presented in Table 1.

## 4. Discussion

Studying the genetic and physiological basis of N uptake and N utilization efficiency in maize could help to improve our understanding of how these two processes contribute to the agronomic performance of maize. These studies could further improve N use efficiency in this crop, which is of major economic importance worldwide [35]. Genetic variability in maize still remains to be fully exploited, in particular when there is a shortage of N during key steps in the developmental cycle, such as the post-silking period. In this investigation, using a core collection of 15 maize inbred lines representative of the genetic diversity of both America and Europe, we examined the impact of N limiting conditions on N uptake and N remobilization in maize, following the silking process.

At the beginning of stem elongation, maize plants were labeled with $^{15}NO_3^-$ in order to estimate the proportion of N remobilized from the vegetative tissues by measuring the $^{15}N$ content in the kernels. In a previous study, it has been shown that $^{15}N$-labeling at the beginning of rapid stem growth appears to be a useful tool for investigating the genetic variability of N remobilization using a large number of genotypes, as the proportion of N taken up after silking significantly contributed to the N budget of the whole plant [32]. Under agronomic conditions, residual N present in the soil could interfere with the estimation of the amount of N remobilized after silking [32]; therefore, plants were grown in large pots in a glasshouse and watered with a nutrient solution. Such semi-controlled growth conditions, allowed a good development of the roots both during the vegetative phase under N$^+$ conditions and following the post-silking N-limiting period. In agreement with this, we observed

that in the glasshouse, KY was on average 40% higher compared to plants grown in the field. This is likely due to more favorable environmental conditions, including temperature and mineral nutrient availability, which have a greater effect on the lines of tropical origin.

Total plant N at maturity (shoots + kernels) was also measured in order to quantify the amount of N taken up by the plants and whether N limiting conditions after silking had any impact on the total plant N content at maturity. For most lines, N limiting conditions after silking had a negative impact on whole plant N uptake (30% decrease on average), except in lines ND283 and Lo32. Similar amounts of N were taken by these two lines⁻ until maturity and there was no marked impact on KY. Kernel yield, represents sink-strength and was much lower in ND283 and Lo32 compared to the other lines (Figure 1). Such a finding suggests that the two lines were able to take up enough N before the silking period, and were, in turn, more tolerant to a post-silking N-deficiency. It has been previously reported that inbred lines can be tolerant to N-deficiency as their KY remains practically unchanged under N limiting conditions, which indicates that they have a higher NUE [8]. In contrast, in a survey describing the adaptation of maize to low N environments, it was concluded that such an adaptation was due to the ability of modern hybrids to accumulate more N after silking while maintaining their productivity in terms of KY [36]. Such results confirm that compared to lines, hybrids are able to take up more N after silking [32]. Whether, the ability of lines ND 283 and Lo32 to take up enough N before silking is due to the root architecture [37] or to a higher efficiency of the inorganic N transport system [38] is currently under investigation.

An interesting result was the finding that N-limiting conditions during the post-silking period had, for most of the lines used in the present investigation, no impact on shoot biomass production. In maize hybrids, the occurrence of a linear relationship between post silking N uptake and the stover dry weight was previously reported. However, such a positive relationship was found only in old hybrids released before 1991 [36]. We did not observe such a genotypic-dependent control of dry matter accumulation and partitioning during the grain filling period. This is likely because lines produce, in general, less biomass compared to hybrids, and because, even if post-silking N uptake is reduced, there is still enough N to sustain shoot growth and development until maturity.

When examining plant performance in terms of KY and kernel N content, four main groups were identified. The first was represented by five lines belonging to the Tropical, European, Northern Flint and Corn Belt Dent genetic groups, in which no changes in KY and kernel N content were observed. The finding that both KY and kernel quality remained unchanged can be explained by the fact that more N was remobilized after silking in order to compensate for the shortage in N during the grain filling period. Therefore, we have classified these lines as being tolerant to a post-silking N stress (Figure 1).

In the second group, lines from four different genetic groups (Tropical, European, Northern Flint and Corn Belt Dent), were also present. Kernel Yield was not reduced, and three of them (P465P, C105, Mo17) were among the most highly productive, although there was a decrease in kernel N content. Two lines belonging to group 2 (P465P and C105) were characterized by an increase in N remobilization. However, in these two lines, such an increase was presumably not sufficient to compensate for the decrease in kernel N content resulting from the lack of N after silking. However, as KY was not reduced, the four lines belonging to group 2 were classified as semi-tolerant to a post-silking N stress.

In the third group of lines represented mostly by Corn Belt Dent lines (4 out of 6 lines in total), N-limiting conditions after silking led to a decrease in KY without any marked changes in the kernel N content. In two lines belonging to this group (FV252 a corn Belt Dent line and ND36 a Northern Flint line), N remobilization was higher, but apparently was not able to compensate for the decrease in KY. However, unlike two lines of group 2 (P465P and C105), the kernel N content remained similar to that of plants grown under non-limiting post-silking N conditions.

Group 4 was represented only by the Europen Flint line FV2 (Table 1). In this line, under N-conditions, there was a marked decrease in KY (more than 3-fold), which moved the ranking in terms of yield from a high yielding line to a low yielding line. Line FV2 was also unique in terms of

response to post-silking N-limiting conditions, as both N uptake and N remobilization were lower, while kernel N content was higher. Such an increase in kernel N content can be explained by the strong sink strength limitation compared to most of the other lines (four-fold reduction in KN). The N concentration increased in the reproductive organs as the amount of available N in the shoots (although reduced in N$^-$) remained relatively high (Table 1). A similar pattern of N management strategy was observed for line B73, except that under our experimental conditions, the increase in kernel N content (%NK), although visible, was not significant. Such findings are in agreement with those of Rajcan and Tollenaar 1999 [39], who showed that the proportion of N in the maize grain can be variable, depending on post-silking N uptake and on the source: sink ratio.

In most of the studies aimed at investigating the effect of N application rates on KY and kernel N content, only a limited number of genotypes (mostly represented by hybrids) have been studied. In some cases, it has been observed that the kernel N content was higher when more N was remobilized from the shoots [40]. In other studies, it was proposed that enhancing post-silking N uptake, rather than N remobilization, was a possible way to increase kernel N accumulation [41]. When QTLs for post-silking N uptake were investigated, only low genetic variations for this trait were generally observed, and thus, a low number of chromosomal regions involved in the control of this trait were detected [42]. Such findings are consistent with the results obtained in the present study, as line FV2, which was used to produce the inbred line population for the detection of QTLs for post-silking N uptake, is one of the least efficient lines in terms of total plant N uptake either in N$^+$ or in N$^-$ (Table 1, Figure 1).

Although it has been shown that N$^-$ remobilization can be maximized if a large amount of N is accumulated before silking, there was no correlation between N accumulation before silking and the amount of N remobilized from the shoots to the kernels, under either N$^+$ or in N$^-$ conditions ($r = 0.31$, $p = 0.26$ and $r = 0.16$, $p = 0.57$ respectively). This finding indicates that such a positive correlation between these traits, and one that is representative of N management, is not necessarily found when considering genetically-distant maize lines.

It can therefore be concluded that for metabolic traits [26] in maize lines exhibiting a large genetic variation, N management strategies are much more diverse compared to those found in genotypes originating from a closely-related genetic background and selected in specific areas of the world. Such a large genetic diversity can be exploited irrespective of plant female flowering date and of KY potential, as late-flowering, high yielding lines are able to maintain their agronomic performance even when there is a post-silking N deficiency stress.

## 5. Conclusions

Exploiting more extensive maize genetic diversity using lines of different geographical origin appears to be a promising way to select lines and then to produce hybrids that are able to maintain high agronomic performance, notably when less N is available during the post-silking period. Although such an exploitation of maize genetic resources could be limited by the fact that a number of these lines may be adapted to specific environmental conditions either in tropical or in temperate regions, it will help to improve our understanding of how these lines are able to maintain high yields under low N conditions. Such knowledge, combined with the benefit of modern genetic techniques, could be used for future breeding strategies, which up to now have generally been conducted under high N fertilization input [5], using genotypes originating from specific areas in the world.

**Author Contributions:** I.Q. and B.H. conceived and designed the experiments; I.Q. and C.D.-G. performed the experiments; I.Q. and B.H. analyzed the data; C.D.-G. contributed reagents/materials/analysis tools; B.H. and P.J.L. wrote the paper.

**Acknowledgments:** This study was funded by INRA (Institut National de la Recherche Agronomique), which also covers the cost of publishing in open access. We also thank Anne Marmagne and Michel Lebrusq for excellent technical assistance in $^{15}$N analyses and in growing the plants, respectively.

**Conflicts of Interest:** The authors declare no conflict of interest.

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
