# Peer review of "Variability for Nitrogen Management in Genetically-Distant Maize (Zea mays L.) Lines: Impact of Post-Silking Nitrogen Limiting Conditions"

_agronomy, doi:10.3390/agronomy8120309_

Reviewer 1 Report

The manuscript is interesting to read. However, it needs to be improved further for publication. Authors needs to provide more information about their experimental design and statistical analysis. Authors mentioned that it is split plot in randomized complete block, but they did not conduct any anova testing about the main effects and their interactions. All we are seeing is the t-test between N treatments for individual variety. Since, it is glasshouse experiment, I think it need to be repeated once again before publishing. Specific comments are provided in the attached file.

Author Response

For clarity we have indicated the changes in red font for Reviewer 1 and blue font for Reviewer 2 in the revised version R1 of the manuscript. The deleted sections are indicated with crossed out text.

 Response to Reviewer 1 (tracked changes indicated in red font)

The manuscript is interesting to read. However, it needs to be improved further for publication. Authors needs to provide more information about their experimental design and statistical analysis. Authors mentioned that it is split plot in randomized complete block, but they did not conduct any anova testing about the main effects and their interactions. All we are seeing is the t-test between N treatments for individual variety. Since, it is glasshouse experiment, I think it need to be repeated once again before publishing. Specific comments are provided in the attached file.

We have provided some additional information on the experimental design and performed ANOVA testing. These can be found below in the responses to the different comments.

We agree that ideally it would have been better to repeat the experiment. However, we consider that the growth conditions were carefully controlled in the glasshouse (notably N feeding conditions, all mineral nutrients, watering and temperature with a cooling system) and that kernel yield was similar or even better for the tropical lines compared to that which was obtained in the field. In the majority of our previously published work, we have presented data from maize (including the lines used in the present study) that has only been carried out during one growing season.

Moreover, due to the very large genetic diversity of the lines and large differences obtained in the measured traits and their mode of N management, we think that they will not be markedly different in a second year of experimentation.

 Line 16 : reword the sentence and delete we

 The sentence has been modified line 16.

 Line 25: Introduction section can be reduced. apart from last 1 or 2 paragraphs, all other paragraphs are about importance of N in production (we already know this), which can be cut short. Instead, more focus should be on N partitioning within the plants, what we know and what we dont know. this study sounds interesting, but this section needs to be revised.

 The introduction has been reduced and the aim of the study has been emphasized at the end of the section lines 89-93.

 Line 105: Reword as "European flint lines were rom France and Italy, whereas all...north america.

 The sentence has been modified line 108.

 Line 116: please be specific

Four times a day is now indicated on line 119

 Line 117: there should be gap between the number and units. Correct throughout the manuscript

The spacing between numbers and units has been corrected throughout the manuscript.

 Line122: what was the growth stage of the maize plants at this time?

Now indicated line 125.

 Line 130: if it was a split plot in randomized complete block design..please clearly define what was the main plot factor and split plot factor for this study? what was the size of pots used in the study and any other related details.

 what was fixed and random effect factor? how many replications?

 The description of the experimental design was not correct and we apologize for this. The right description and additional information concerning the experimental design is now presented in lines 128 to 131.

 Line 133: this should be results section, not in M & M section

 This part of the Material and Methods section has been moved to the Results section lines 176 to 184.

 Line 143: what is meant by this? what growth stage?

 The sentence has been modified line 140

 Line 153: what are these plants? No N added?

It is now indicated that these were unlabeled control plants in line 150

 Line 166: is there any overall anova testing conducted for this split-plot designed experiment. To my knowledge, the t-test only give you comparison between treatment means only. Question, is there effects of varieties, N treatment and their interaction on Ky and other parameters measured in the study.

 When a two-way ANOVA was performed, we found that for all the measured traits there was a significant genotypic effect (P<0.0001). However, this result cannot be fully exploited due to the large genetic diversity of the lines (see ref 26), which does not provide any major additional information with respect to the impact of post-silking N nutrition on plant yield and plant N management.

Nevertheless, we have included this statistical analysis in the Results section lines 186-187

 Line 175: no need to repeat this information again here. please delete

Deleted, lines 172-176

 Line 195: Please read and correct the sentence

The sentence has been corrected line 203.

 Line 205: read and correct the sentence

Sentence corrected line 210.

 Line 207: unaffected in N-...compared to what? N+ treatments?

The information is now provided line 215.

 Line 210: no need to repeat this information again here. please delete

Deleted

 Line 214: is this %NK or Nt? please use the same terms you used in the tables to be clear.

15% reduction? or 15% was the N take up? Reword the sentence to be clear.

It is Nt, the sentence has been modified line 222.

 Line 215: no need to mention this if it is not significant

Deleted

 Line 223: delete

The sentence has been modified line 230.

 Line 243: not clear the use of this figure. You already presented all the data in the table, no need to present this again in different sets of groupings.

We think that this figure needs to be presented as it gives a comprehensive view of the results obtained in the study along with the agronomic and developmental characteristics of the 15 lines.

 Line 246: if this is part of figure, then it should be in the figure caption, not as part of the manuscript text

This section was part of Figure 1 caption. It has been corrected in the revised version of the manuscript lines 251 to 267.

 Reviewer 2 Report

Minor corrections/suggestions:

 (1) L26. In the current form, the following fragment is incorrect: “…mineral fertilizers such as nitrogen (N)…”. Nitrogen is a chemical element and important component of fertilizers. Please re-phrase or modify.

 (2) L76-77. English needs improvement in the sentence with three times used verb ‘is’: “…the grain N which IS a source of proteins for both human and animals, IS provided from pre-existing N in the stover before silking, a process that IS strongly…”.

 (3) L94. Please insert important term ‘inbred’ making the studied material as ‘inbred lines’. I checked Reference [27], where exactly ‘inbred lines’ were described. Regardless other published papers, Authors have to be careful with the term ‘line’ or ‘lines’ as unclear one. Its maybe ‘breeding lines’, ‘recombinant inbred lines’, ‘hybrid lines’ or other.

 (4) L99. Please use Reference in square brackets [27] instead year in the phrase: “…by Camus-Kulandaivelu et al. [27]…”.

 (5) L102. The indicated Reference [29] has been used incorrectly. It does not contain any information about SSR or SNP markers. Instead, Reference [27] is correct for SSR, and Authors have to check and insert correct Reference for SNP markers.

 (6) L243-245. Legend of Figure 1 needs full descriptions of all abbreviated names and units, including last three columns, KYL-N+; KYL-N-; Sd. Authors have to extract this information from the following paragraph and insert it in the Figure Legend.

 (7) L276 and L111. The conflicting information is present in M&M section and discussed. Pots with “…diameter and height of 30 cm…” (L111) are not so big as mentioned that plants “…were grown in large pots…” (L276). For Arabidopsis such pot size is large, indeed, but its maybe moderate size of pots for wheat or barley. However, for such crops as maize, pots with 30 cm diameter and 30 cm height are too small. Growth cylinders with 50 cm diameter and 1 m height would be much better for maize. I would suggest Authors to insert one sentence with their explanation in the Discussion section, why they believe that the used size of pots are the most optimal for their experiments.

Author Response

For clarity we have indicated the changes in red font for Reviewer 1 and blue font for Reviewer 2 in the revised version R1 of the manuscript. The deleted sections are indicated with crossed out text.

 Response to Reviewer 2 (tracked changes in blue font)

 Comments and Suggestions for Authors

Minor corrections/suggestions:

 (1)           L26. In the current form, the following fragment is incorrect: “…mineral fertilizers such as nitrogen (N)…”. Nitrogen is a chemical element and important component of fertilizers. Please re-phrase or modify.

The sentence has been modified line 26.

(2) L76-77. English needs improvement in the sentence with three times used verb ‘is’: “…the grain N which IS a source of proteins for both human and animals, IS provided from pre-existing N in the stover before silking, a process that IS strongly…”.

The sentences have been modified, lines 75 to 77.

(3) L94. Please insert important term ‘inbred’ making the studied material as ‘inbred lines’. I checked Reference [27], where exactly ‘inbred lines’ were described. Regardless other published papers, Authors have to be careful with the term ‘line’ or ‘lines’ as unclear one. Its maybe ‘breeding lines’, ‘recombinant inbred lines’, ‘hybrid lines’ or other.

Inbred is now indicated in lines 96, 98, and in the title of Figure 1, line 251.

 (4) L99. Please use Reference in square brackets [27] instead year in the phrase: “…by Camus-Kulandaivelu et al. [27]…”.

Modified line 101.

 (5) L102. The indicated Reference [29] has been used incorrectly. It does not contain any information about SSR or SNP markers. Instead, Reference [27] is correct for SSR, and Authors have to check and insert correct Reference for SNP markers.

We have added reference 27 line 104. However, we think that ref 29, still needs to be cited because in this paper a Table summarizing the genotyping of the lines is presented.

 (6) L243-245. Legend of Figure 1 needs full descriptions of all abbreviated names and units, including last three columns, KYL-N+; KYL-N-; Sd. Authors have to extract this information from the following paragraph and insert it in the Figure Legend.

This information is presented at the end of the reformatted legend of Figure 1. Lines 265-267

 (7) L276 and L111. The conflicting information is present in M&M section and discussed. Pots with “…diameter and height of 30 cm…” (L111) are not so big as mentioned that plants “…were grown in large pots…” (L276). For Arabidopsis such pot size is large, indeed, but its maybe moderate size of pots for wheat or barley. However, for such crops as maize, pots with 30 cm diameter and 30 cm height are too small. Growth cylinders with 50 cm diameter and 1 m height would be much better for maize. I would suggest Authors to insert one sentence with their explanation in the Discussion section, why they believe that the used size of pots are the most optimal for their experiments.

The exact size and volume of the pots are now indicated on line 114.

The results presenting the comparison between the agronomic performances of the plants between the field and the glasshouse is now presented at the beginning of the results section ( as requested by reviewer 1). A sentence explaining that the plants growth conditions were optimal in the glasshouse has been added on line 184 to 185.